# Microwave-assisted synthesis of metal-organic chalcogenolate assemblies as electrocatalysts for syngas production

Hannah Rabl [1], Stephen Nagaraju Myakala[1], Jakob Rath [1], Bernhard Fickl [1], Jasmin S. Schubert [1], Dogukan H. Apaydin [1✉] & Dominik Eder[1]

Today, many essential industrial processes depend on syngas. Due to a high energy demand and overall cost as well as a dependence on natural gas as its precursor, alternative routes to produce this valuable mixture of hydrogen and carbon monoxide are urgently needed. Electrochemical syngas production via two competing processes, namely carbon dioxide ($CO_2$) reduction and hydrogen ($H_2$) evolution, is a promising method. Often, noble metal catalysts such as gold or silver are used, but those metals are costly and have limited availability. Here, we show that metal-organic chalcogenolate assemblies (MOCHAs) combine several properties of successful electrocatalysts. We report a scalable microwave-assisted synthesis method for highly crystalline MOCHAs ([AgXPh] $_\infty$: X = Se, S) with high yields. The morphology, crystallinity, chemical and structural stability are thoroughly studied. We investigate tuneable syngas production via electrocatalytic $CO_2$ reduction and find the MOCHAs show a maximum Faraday efficiency (FE) of 55 and 45% for the production of carbon monoxide and hydrogen, respectively.

[1] Institute of Materials Chemistry, TU Wien, Getreidemarkt 9, 1060 Vienna, Austria. ✉email: dogukan.apaydin@tuwien.ac.at

Syngas is a widely used gaseous mixture of carbon monoxide and hydrogen. Today´s population is highly dependent on syngas as it is used as a precursor to many essential chemicals, including bulk chemical intermediates such as methanol, thus being relevant for the production of polymers and pharmaceuticals[1–4]. Thanks to the development of Fischer–Tropsch process, syngas can be considered a potential source of synfuels and synthetic petroleum[5–7]. Furthermore, syngas is a valuable good due to its high hydrogen content. Hydrogen is known for its good power generation capabilities[2,8]. It is advantageous, as it provides the highest energy per unit mass of all known fuels and does not lead to $CO_2$ emissions upon combustion[3,9]. Another use of hydrogen is its capability in saturating double bonds and acting as a reducing agent[2]. Hydrogen is also essential in the Haber Bosch process for ammonia synthesis[2,10]. Syngas is, therefore indispensable for the world's food supply, as approximately 88% of ammonia is used in fertilizers[10]. Nowadays, Syngas is mainly produced via three processes: steam reforming[2,11–13], partial oxidation[2], or thermal decomposition[3]. All three processes however require high temperatures and pressures, fossil fuel precursors, and rare and expensive metal catalysts[2,3,12–14]. Thus, alternative synthesis routes for syngas production are of great need. Besides using biomass[14–16], thermocycles[1,17–21] or photocatalysis[1,22,23], electrochemical $CO_2$ reduction to CO and $H_2$ is a promising syngas production method[1,24–29]. So far various metals, alloys, metal oxides, and metal-chalcogenide catalysts have been tested[30–47]. Silver shows high selectivity towards CO due to its lower binding energy towards CO intermediate, making a further reduction to hydrocarbons hardly possible[31]. Silver-chalcogenides, including $Ag_2Se$ and $Ag_2S$ were recently reported as highly efficient catalysts for carbon dioxide reduction reaction ($CO_2RR$)[48,49]. Optimization of selectivity, efficiency and cost of suitable catalysts for electrochemical $CO_2RR$ and syngas production is still a huge challenge in the field[30,36].

In 2002 Cuthbert et al. reported the space group and structural alignment of an organic-inorganic 2D hybrid material consisting of silver metal centres with coordination to a chalcogenide covalently binding to phenyl rings[50]. This coordination polymer was named *Mithrene* ([AgSePh]$_\infty$) by the group of Nathan Hohman in 2018. Since then, the Hohman group has done extensive research on the synthesis, characterization and the properties of these metal organic chalcogenolate assemblies (MOCHAs)[51–59]. In 2020 they introduced the so-called tarnishing approach and a biphasic synthesis of MOCHAs[52,53,60]. Tarnishing is based on the reaction of a diphenyl dichalcogenide precursor with a metal substrate at slightly elevated temperatures. This method requires highly pure metal substrates or metal films as precursors. A MOCHA film bound to the substrate obtained via this method limits the use of various substrates for growing MOCHA films. In addition, such metal thin films require a physical vapour deposition setup which is not a common laboratory infrastructure. Furthermore, it was shown that the reaction of diphenyl dichalcogenides with silver oxide did not lead to MOCHA formation. Silver oxide formed MOCHAs only with benzeneselenol or -thiol precursors. These reactants however, are highly toxic and thus their handling and utilization should be kept to a minimum. Crystallographic and optical studies on MOCHAs in particular on the silver benzeneselenolate MOCHA "*Mithrene*" have been carried out since then[51,53–55,57,61–64]. However, either the quantities are very low (biphasic synthesis) or MOCHA film thickness (tarnishing method) does not exceed 20 nm. Such limitations seem to hinder further applications of MOCHAs.

We developed a microwave-assisted process for metal organic chalcogenolate assemblies that is capable to shorten the synthesis time from 3 days to only 1–5 h and increases the absolute yield of the material by a factor of ~100 compared to the biphasic synthesis method suggested by Schriber et al. Furthermore, we evaluated the performance of [AgSePh]$_\infty$ and [AgSPh]$_\infty$ for electrocatalytic syngas production. In addition to catalytic studies, we perform long-term stability studies of MOCHAs in various environments.

## Results and discussion

**Synthesis and structural characterization of MOCHAs.** So far, different synthesis methods for MOCHAs did not result in high yields, which hinder further use of MOCHAs. Here we report a new and more advantageous approach for MOCHA synthesis with a microwave reactor using a suspension. Consequently, product isolation is easily achieved by filtration and consecutive washing and drying of the powder. Furthermore, we were able to achieve yields up to 83% (taking a molecular weight of 264 g mol$^{-1}$ for [AgSePh]$_\infty$, as suggested by Schriber et al.[51]), where the absolute amount obtained (110 mg) is only limited by the reactor size. Another advantage of this routes is that the microwave synthesis does not require the use of toxic and smelly precursors (e.g., thiophenol, benzeneselenol)[51] or the use of high purity metals[52].

X-ray diffraction (XRD) and scanning electron microscopy (SEM) confirms the successful synthesis of *Mithrene* ([AgSePh]$_\infty$) and *Thiorene* ([AgSPh]$_\infty$) (Fig. 1a, b). Note that the XRD pattern exhibits three characteristic peaks corresponding to the {002}, {004}, and {006} planes, respectively[51,59]. We were interested in reducing the synthesis time, which is why we performed MOCHA synthesis via a microwave assisted method for 1, 3, 5, and 14 h. We obtain the same characteristic pattern using microwave synthesis at 110 °C with all tested durations (Fig. 1c). We also employed hydrothermal synthesis at 150 °C as an alternative synthetic route. However, hydrothermal synthesis yielded products only after 72 h, manifesting again the advantage of microwave synthesis, where only 1 h was required. This suggests that the reaction between $AgNO_3$ and diphenyldichalcogene (the precursors) has thermal/kinetic limitations. Microwave synthesis is, in general, known to increase the reaction rate, which helps to break the limitations by introducing a more uniform heating throughout the reaction vessel, decreasing the reaction time tremendously[65]. ATR-FTIR measurements can help to reveal eventual impurities. An incomplete reaction, for instance, would result in a prominent band at around 2305 cm$^{-1}$, corresponding to the benzeneselenol or benzenethiol species. As we do not observe any bands at 2305 cm$^{-1}$ (Fig. S1) we conclude that the powders contain only the products. Both MOCHA samples display similar vibrations as they can be attributed mainly to the aromatic network in [AgSePh]$_\infty$ and [AgSPh]$_\infty$. At 1434, 1471, and 1572 cm$^{-1}$, aromatic C–C stretches are observed. Vibrations at 995, 1018, and 1067 cm$^{-1}$ correspond to C–H in plane stretching while C–H aromatic out of plane vibrations can be seen at lower wavenumbers around 615, 688, 725, and 738 cm$^{-1}$. Our measurements are in good agreement with literature spectra[66].

The band gap energies of [AgSePh]$_\infty$ and [AgSPh]$_\infty$ were measured via UV-Vis-DRS (Fig. 1d) and calculated as 2.7 and 3.4 eV, respectively. The difference in bandgap can be traced back to the structural difference between [AgSePh]$_\infty$ and [AgSPh]$_\infty$. In [AgSePh]$_\infty$ Ag–Ag network is trigonal, whereas in [AgSPh]$_\infty$ the Ag-Ag network is part of a linear chain[51]. Thus, excitons are eventually delocalized in two dimensions above and below the silver network in [AgSePh]$_\infty$, causing absorption and emissive behaviour. [AgSPh]$_\infty$ on the other hand, does not exhibit a delocalization of excitons in two dimensions due to Ag-Ag linear chains, causing a lack of emission in the visible range for [AgSPh]$_\infty$.

Further characterization results, including energy dispersive X-ray (EDX) (Fig. S2) and X-ray photoelectron spectroscopy

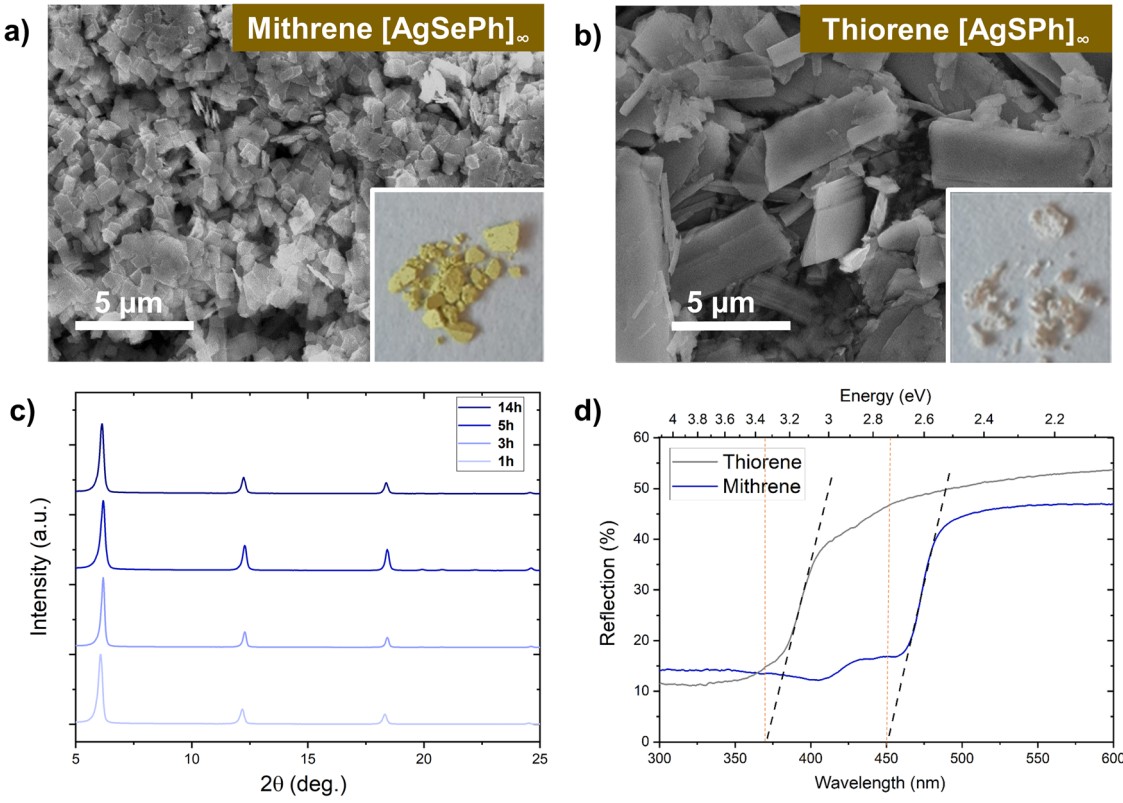

**Fig. 1 Characterization of MOCHAs [AgSePh]∞ and [AgSPh]∞ by means of SEM and XRD after performed microwave synthesis. a** SEM image of [AgSePh]∞ MOCHA, **b** SEM image of [AgSPh]∞ MOCHA. **c** XRD pattern of [AgSePh]∞ after various synthesis durations. **d** UV–Vis-DRS spectra of [AgSePh]∞ and [AgSPh]∞.

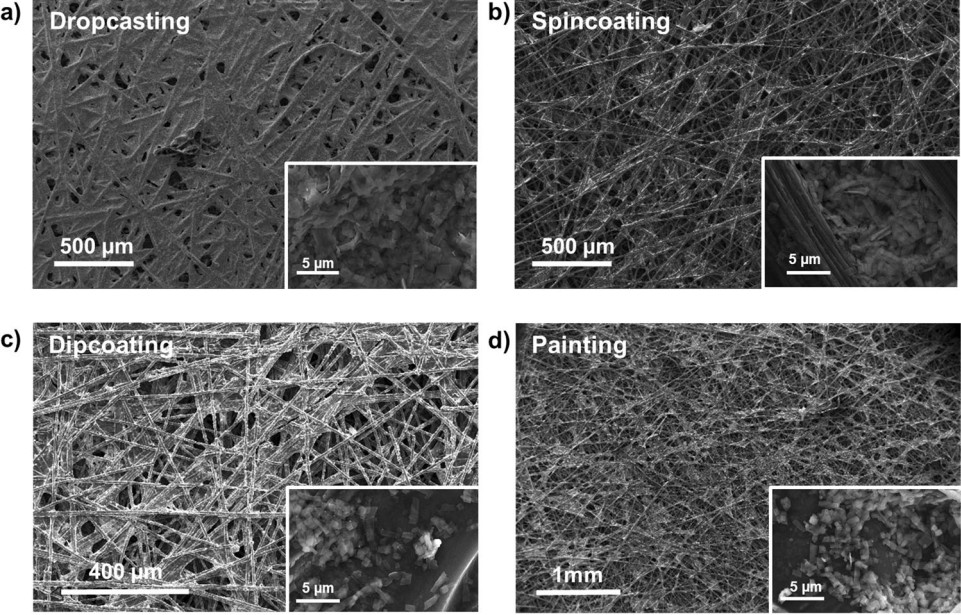

**Fig. 2 SEM images of deposited [AgSePh]∞ on carbon paper (CP).** Deposition obtained via **a** dropcasting, **b** spin coating, **c** dipcoating and **d** painting.

(XPS) (Fig. S3) data can be found in the supplementary information in Supplementary Table 1. These measurements focussed on the determination of elemental ratio in MOCHAs. EDX measurements lead to overestimation of carbon content by 10% (atomic) and underestimated the metal to chalcogenide ratio. XPS measurements of both MOCHAs showed results that are in good agreement to theoretical calculations.

*Electrode preparation.* The MOCHAs were deposited on carbon paper comparing various techniques, including drop-casting, spin-coating, dip-coating, and simple painting. The effect of the deposition technique on the substrate coverage was investigated using SEM. As seen in Fig. 2, drop-casting led to a full coverage of CP, spin-coating gave a rather poor coverage. Painting and dip-coating resulted in comparable amounts of MOCHA on CP.

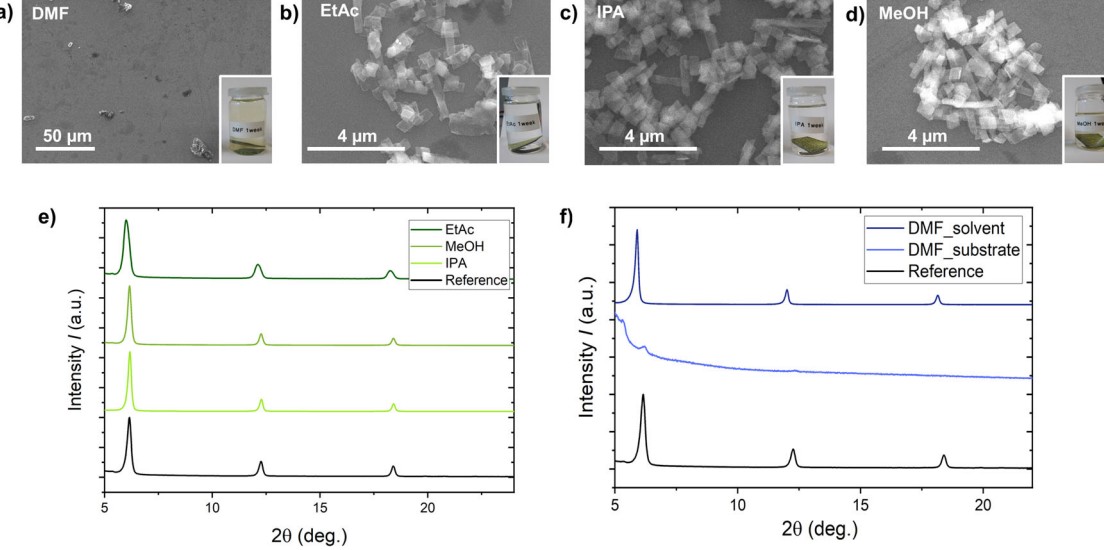

**Fig. 3 Stability test results performed on [AgSePh]$_\infty$ in various solvents at room temperature for one week. a–d** SEM images of Si-substrates with [AgSePh]$_\infty$ MOCHA after stability testing. XRD pattern of **e** substrate+MOCHA after testing in EtAc, IPA, MeOH, and **f** of the substrate and solution after testing in DMF.

However, dip-coating yielded more reproducible results compared to painting. Hence, dip-coating was chosen as our standard approach for electrode preparation.

*Chemical and structural stability.* MOCHAs are chemically and structurally stable, when stored in dark at room temperature[52,53]. We further investigated their stability in harsher environments/ conditions, using [AgSePh]$_\infty$ films deposited onto a Si substrate (for ease of characterization) and polycrystalline [AgSePh]$_\infty$ powder. For the sake of brevity, we only display the results of the stability tests in organic solvents (Fig. 3). Results of further stability tests can be found in Supplementary Information (Figs. S4–S6). Variations of pH exposure showed no change in structural or morphological appearance of [AgSePh]$_\infty$ from pH 0 to 9. The size and shape of [AgSePh]$_\infty$ crystals remained unchanged, and the powder XRD pattern displayed the three characteristic peaks. We conclude that [AgSePh]$_\infty$ is stable in acidic and basic conditions. Exposure of [AgSePh]$_\infty$ to various temperatures in the range of −25 to 120 °C was investigated as well. The results are shown in Fig. S5. We do not observe any change in shape, morphology, or size of the crystals by means of SEM. Also, the XRD pattern confirmed no structural changes to the unexposed [AgSePh]$_\infty$ pattern.

Stability tests upon light illumination with different wavelengths (365 nm, 456 nm, Solar Simulator) showed that [AgSePh]$_\infty$ powder turned from a bright yellow colour before illumination into a darker brownish-yellow powder upon illumination times longer than 24 h. (Illumination was performed from 10 min until 72 h). This colour change was retained even after months of storing. However, no changes were observed in XRD. Thus, we can conclude that the synthesized MOCHAs are highly stable upon light illumination.

Initial inspection of the Si substrate immersed in DMF showed no traces of [AgSePh]$_\infty$ (see Fig. 3a). However, it was noted that the DMF solution turned yellowish. After evaporation of the DMF, a yellow residue remained. XRD pattern of this residue showed the three characteristic peaks of [AgSePh]$_\infty$ corresponding to the {002}, {004}, and {006} planes. This implies, DMF does not alter the chemical and structural composition of [AgSePh]$_\infty$. DMF leads only to lifting-off of [AgSePh]$_\infty$ from the surface.

In the other tested solvents (ethyl acetate, isopropanol and methanol) No lift-off of [AgSePh]$_\infty$ could be seen even after being

immersed for one week (Fig. 3b–d) and their stability is confirmed by the measured XRD pattern (Fig. 3e).

*Electrocatalytic performance.* Electrocatalytic $CO_2$ reduction was performed at three different potentials (−0.6, −0.8, and −1.0 V vs. RHE) for 30 min. Each experiment was done a minimum of two times, with their average values shown in Fig. 4. The products were analyzed by means of gas chromatography. The faradaic efficiency (FE) and rate of the products CO and $H_2$ is given for these electrodes: bare carbon paper, [AgSePh]$_\infty$ on carbon paper, [AgSPh]$_\infty$ on carbon paper, (CP, [AgSePh]$_\infty$/CP and [AgSPh]$_\infty$/CP) in Fig. 4.

The bare CP electrode at all potentials produced only $H_2$. For MOCHA/CP electrodes, the combined product formation (CO and $H_2$) also decreased with decreasing potential, due to the lower energy input and thus less help in overcoming the activation barrier. However, a distinct behaviour can be noted when comparing the ratio of CO to $H_2$. While both $FE_{CO}$ and CO rate decreased with decreasing potentials, $FE_{H2}$ increased and $H_2$ rate decreased. Thus, the CO:$H_2$ ratio was in favour of CO @−1.0 V vs. RHE, and in favour of hydrogen at potentials lower or equal to −0.8 V vs. RHE. At −1.0 V vs. RHE both MOCHA electrodes achieved 55% $FE_{CO}$. Their rates, however, showed distinct behaviour. A rate of 61 μmol h$^{-1}$ CO was obtained with [AgSePh]$_\infty$/CP, while [AgSPh]$_\infty$/CP achieved 85 μmol h$^{-1}$ CO. Faraday efficiency and rate for carbon monoxide were clearly higher in the case of [AgSPh]$_\infty$ electrode compared to [AgSePh]$_\infty$ at −0.8 V vs. RHE. With [AgSPh]$_\infty$/CP 37% $FE_{CO}$ and 21 μmol h$^{-1}$ CO were obtained whilst [AgSePh]$_\infty$/CP only reached 6% $FE_{CO}$ and 2 μmol h$^{-1}$. From these observations we conclude that [AgSPh]$_\infty$ can be considered as a more selective catalyst for $CO_2$RR to CO with a higher performance compared to [AgSePh]$_\infty$. On average [AgSPh]$_\infty$ produced more CO per hour (an additional 17 μmol) compared to [AgSePh]$_\infty$. At −0.6 V vs. RHE with both electrodes almost no CO could be detected, and also hydrogen rate was rather low reaching only values up to 40 μmol h$^{-1}$.

Notably, overall Faraday efficiencies lay below 100% at lower overpotentials (−0.6 V). An explanation for that could be an unsaturated electrolyte. Although the solubility of the products hydrogen and carbon monoxide in the electrolyte 0.5 M KHCO$_3$ is low, a saturation of the liquid has to be obtained before the gas phase is filled. When taking a sample from the head space (gas phase) at

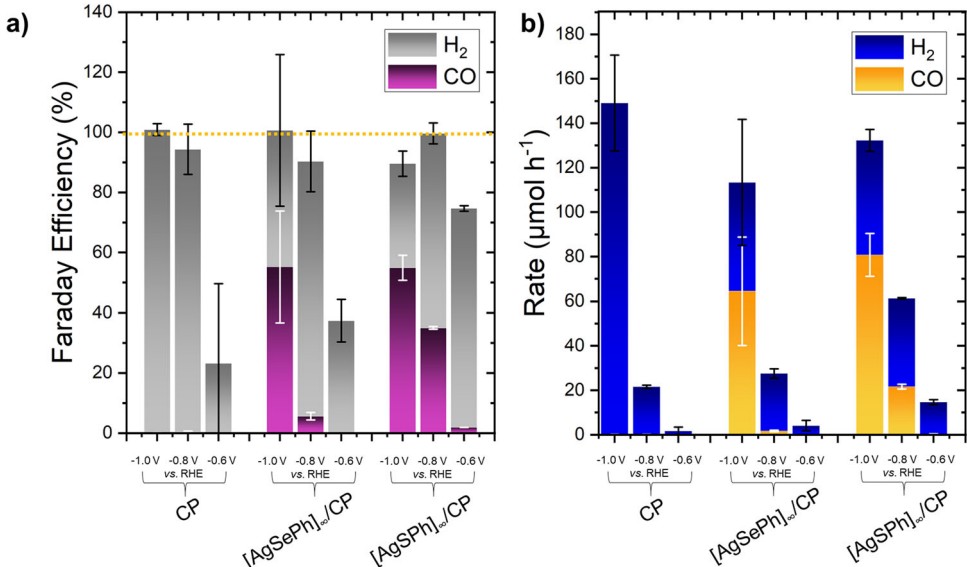

**Fig. 4 Results of electrocatalytic syngas production using MOCHA/CP electrodes. a** Faraday efficiencies and **b** rates for the products CO and $H_2$ obtained with three different electrodes (CP, [AgSePh]/CP, [AgSPh]/CP) at three different potentials (−1.0 V, −0.8 V, −0.6 V vs. RHE). Error bars (black for $H_2$ and white for CO) correspond to the fluctuations of the detected amount of product.

**Table 1 Comparison of various catalyst performances for Syngas production.**

| Catalyst | Potential range (Vvs. RHE) | Current density (mA cm$^{-2}$) | CO:$H_2$ ratio | FE$_{CO}$ (%) | FE$_{Total}$ (%) | Electrolyte | Refs. |
|---|---|---|---|---|---|---|---|
| Pd/C | −0.5 ~ −1.0 | 0.6 (−0.7 V) | 0.5-2.0 | 60 | 90 | 0.5 M NaHCO$_3$ | 72 |
| Zn/Cu foam | −0.6 ~ −1.3 | 40 (−1.3 V) | 0.2-2.31 | 40 | 85 | 0.5 M KHCO$_3$ | 73 |
| Au-NPs | −0.6 ~ −1.3 | / | 0.5-1.0 | 45 | 100 | 0.1 M KHCO$_3$ | 74 |
| Fe/FeN$_4$C | −0.45 ~ −0.8 | 39.33 (−0.8 V) | 0.1-0.9 | 52 | 100 | 0.5 M KHCO$_3$ | 75 |
| MoSeS alloy | −0.6 ~ −1.6 | 43 (−1.15 V) | 1.0 | 45.2 | 96 | EMIM-BF$_4$ solution | 47 |
| This work | | | | | | | |
| [AgSePh]$_∞$ | −0.8 ~ −1.0 | 1.6 − 6.3 | 0-1.2 | 5-55 | 89-100 | 0.5 M KHCO$_3$ | / |
| [AgSPh]$_∞$ | −0.8 ~ −1.0 | 3.3 − 7.9 | 0.5-1.6 | 34-54 | 88-98 | 0.5 M KHCO$_3$ | / |

low potentials, oversaturation in the liquid phase might not have been achieved, leading to a reduced Faraday Efficiency.

Comparing MOCHAs to similar literature examples proved to be a challenging task as such material class has not been shown before for catalysis. We see silver chalcogenides as fitting counterparts for the performance comparison. [AgSPh]$_∞$ was compared to Ag$_2$S on carbonaceous electrode substrates as a reference, which reached FE$_{CO}$ values up to 67%[49]. Ma et al. investigated the difference in CO$_2$ reduction performance between *monoclinic (m)* and *orthorhombic (o)* Ag$_2$Se catalysts[48]. They found, that *m*-Ag$_2$Se was clearly favouring CO formation compared to *o*-Ag$_2$Se. [AgSePh]$_∞$ electrode showed a 15% higher FE$_{CO}$ compared to their *o*-Ag$_2$Se electrode. As [AgSePh]$_∞$ MOCHA, however; crystallizes in a monoclinic space group, comparison to *m*- Ag$_2$Se (FE$_{CO}$ 98%@−0.9 V vs. RHE[48]) is a fair comparison. Silver chalcogenides lead to a high CO:$H_2$ ratio, whereas MOCHAs are able to produce a well-suited ratio of 1:1 for syngas production.

Electrocatalytic CO$_2$RR in aqueous environment is accompanied by the hydrogen evolution reaction (HER) due to the simultaneous reduction of water (or protons H$^+$) and CO$_2$. The following two reactions ocurr:[27]

$$CO_2 + 2H^+ + 2e^- \rightleftarrows CO + H_2O \quad E_0 = -0.53\,V\,vs.\,SHE \quad (1)$$

$$2H^+ + 2e^- \rightleftarrows H_2 \quad E_0 = 0.42\,V\,vs.\,SHE \quad (2)$$

By producing Syngas electrocatalytically, the drawback of a competing reaction can be bypassed and the HER fully integrated in the use. Another advantage of electrocatalytic Syngas production compared to other Syngas production techniques, including solid oxide electrolysis cells (SOECs), steam reforming and thermocycles is its room temperature application[1,2,13,25]. Electrocatalytic syngas production is thus a cost effective, environmentally friendly alternative for producing Syngas. However, the finding and development of cheap and suitable catalysts is still a huge challenge in the field. A summary of potential catalysts and their Faraday Efficiencies, current densities, and experimental parameters is summarized in Table 1. and compared to the results obtained with MOCHA/CP electrodes.

**Recyclability and post catalytic stability**. To further study the stability/evolution of MOCHAs in detail, the electrodes were investigated before and after electrolysis. Dip-coated MOCHA/CP electrodes were sonicated in isopropanol after catalytic testing. The obtained MOCHA residue was examined by SEM and XRD (Supplementary Fig. 7). The measurements reveal that [AgSePh]$_∞$ and [AgSPh]$_∞$ were both present on the carbon paper electrode after electrolysis without any structural changes. Next the Recyclability of [AgSePh]$_∞$/CP electrode was studied. We performed four consequent cycles of electrolysis at −1.0 V vs. RHE.

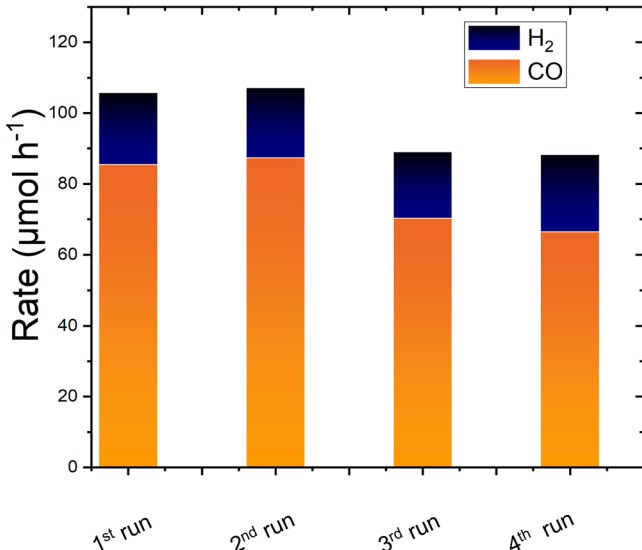

**Fig. 5 Rate of CO and H₂.** Results obtained with a [AgSePh]∞/CP electrode at −1.0 V vs. RHE after using it four times consecutively.

The rates of CO and $H_2$ obtained in each run are summarized in Fig. 5.

The CO formation rate decreased from 85 μmol h$^{-1}$ (1st run) to 67 μmol h$^{-1}$ upon consecutive electrolyses which corresponds to a ~21% decrease in activity. Exact mechanism of deactivation is subject to further studies.

## Conclusion

We developed a new, high-yield microwave assisted synthesis method for Metal Organic Chalcogenolate Assemblies. We report a MOCHA synthesis with a yield of 83 w% where the MOCHA is obtained as a powder, enabling a wide variety of characterization, application and handling possibilities with this 2D material. Compared to previously reported methods, the microwave synthesis can compete in terms of reaction time[63] or even shorten them (comparison to tarnishing[52] and biphasic method[53,59]). Furthermore, we could circumvent the use of toxic (thiophenol, selenophenol) precursors. The synthesis we report here requires no tedious sample separation. Stability measurements indicate that MOCHAs are stable in various solvents, temperature range and pH media as well as upon light irradiation. Electrocatalytic Syngas production was successfully performed with [AgSePh]∞ and [AgSPh]∞ electrodes. Faraday efficiencies and rates up to 55% and 81 μmol h$^{-1}$ CO respectively, show that MOCHAs have a huge potential for electrochemical Syngas production. Further studies will focus on the underlying mechanism and the working principle of MOCHAs in electrocatalytic $CO_2$RR.

## Experimental

Unless stated otherwise all chemicals were used as received.

**MOCHA synthesis.** MOCHAs were fabricated in an Anton Parr Monowave 300 Microwave Synthesis Reactor. 0.5 mmol AgNO₃ (abcr, 99.9%) were dissolved in 2 mL $H_2O$. 0.75 mmol of the diphenyl dichalcogenide (DPDS, DPDSe) (Acros Organics, 99%) were dissolved in 5 mL methanol by sonicating the suspension for 45 min in an ultrasonic bath. These two solutions were combined in the microwave vial which was closed and heat by microwave irradiation at 110 °C for 5 h under continuous stirring at 600 rpm. The obtained MOCHA was rinsed and washed with methanol by vacuum filtration and stored in a drying oven at 40 °C for 24 h.

**Ink preparation.** 100 mg of the MOCHA ([AgSePh]∞ or [AgSPh]∞) were suspended in a mixture of 9.5 mL isopropanol (technical) and 500 μL Nafion™ (Sigma Aldrich, 5 w% in aliphatic alcohols and water) by 30 min sonication.

**Electrode preparation.** Toray Carbon Paper (Thermo Fisher Scientific TGP-H-60, 45356.KS) was cut into 2 × 1 cm pieces and used without further treatment. The CP was dip coated with the "ink" for 24 h. After removing from the ink the electrodes were rinsed with isopropanol (technical) and dried.

**Electrochemistry.** Electrochemical measurements were performed in a custom-designed H-cell (Perfectlight). 0.5 M KHCO₃ was used as an electrolyte for all electrochemical measurements. The electrolyte was sparged with $N_2$ or $CO_2$ until saturation before usage. The cathodic side was equipped with a stirring bar and the working electrode (CP, [AgSePh]∞/CP, [AgSPh]∞/CP) and the headspace was purged for an additional 30 min with $CO_2$ before electrolysis. Ag/AgCl in 3 M KCl was used as the reference electrode whereas a platinum electrode served as the counter electrode. For electrolysis a constant potential of −1.0, −0.8, −0.6 V vs. RHE was applied for 30 min.

**Stability testing.** The chosen irradiation wavelengths and lamps were a Thorlabs UV-lamp (365 nm), a Kessil (PR160) 456 nm lamp and a solar simulator. Stability tests at different temperatures and illuminations were carried out on powder samples of [AgSePh]∞ placed in a glass vial. Testing in different organic solvents and pH media was done on [AgSePh]∞ films. The films were obtained via ink preparation (see 4.2 "Ink preparation") followed by drop-casting the ink onto a Si (711) substrate.

**ATR-FTIR measurements.** ATR-FTIR spectra were taken on a Bruker Tensor 27 instrument in the range of 4000–600 cm$^{-1}$. Prior a background spectrum of air was taken and directly subtracted from the measurement spectrum.

**DRS measurements.** DRS measurements were recorded on a JASCO V-670 Spectrophotometer in a wavelength range from 200 to 900 nm. MgSO₄ (Sigma Aldrich, >99.5%) was used as baseline.

**SEM/EDX measurements.** SEM and EDX images and spectra were taken on a FEI Quanta FEG 250. SEM measurements were performed with an acceleration voltage of 10 kV and a working distance of 10 mm. For EDX measurements an *Octane Elite Super* detector was used. Spectra were recorded using an acceleration voltage of 20 kV, a resolution of 125.2 eV and a take-off angle of 31.7°.

**XPS measurements.** All measurements were carried out on a custom-built SPECS XPS-spectrometer equipped with a monochromatised Al-Kα X-ray source (μFocus 350) and a hemispherical WAL-150 analyzer (acceptance angle: 60°). All samples were mounted onto the sample holder using double-sided carbon tape. Pass energies of 100 eV and energy resolutions of 1 eV were used for the survey (excitation energy: 1486.6 eV, beam energy and spot size: 70 W onto 400 μm, angle: 51° to sample surface normal, base pressure: $5 \times 10^{-10}$ mbar, pressure during measurements: $2 \times 10^{-9}$ mbar). Data analysis was performed using CASA XPS software, employing transmission corrections (as per the instrument vendor's specifications), Shirley/Tougaard backgrounds[67,68] and Scofield sensitivity factors[69]. Charge correction was applied so the adventitious carbon peak (C–C peak) was shifted to 284.8 eV binding energy (BE). All content values

shown are in units of relative atomic percent (at%), where the detection limit in survey measurements usually lies around 0.1–1 at%, depending on the element. The accuracy of XPS measurements is around 10–20% of the values shown. Assignment of different components was primarily done using Reference databases[70,71].

**XRD measurements.** Powder XRD pattern were recorded on a PANalytical X´Pert Pro multi-purpose diffractometer (MPD) in Bragg Brentano geometry. The 2θ ranges were set between 5 and 90 degrees and a scan rate of 4° min$^{-1}$ was applied. The anode material was copper, leading to the emission of $Cu_{K\alpha}$ and $Cu_{K\beta}$ radiation (ratio 2:1) with a wavelength of 1.5406 Å.

## Data availability

The datasets generated during and/or analyzed during the current study are available from the corresponding author on reasonable request.

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

## Acknowledgements
We are grateful to the AIC, USTEM, and X-Ray Centre of TU Wien for providing their facilities. The authors acknowledge TU Wien Bibliothek for financial support through its Open Access Funding Program.

## Author contributions
H.R. synthesized the compounds, performed the electrolyses, performed structural characterization, analyzed the catalytic performance and wrote the manuscript. S.N.M. performed SEM and EDX measurements. J.R., B.F., and J.S.S. performed XPS measurements. D.H.A. conceptualized and supervised the research and wrote the manuscript together with H.R. using the comments from all co-authors. D.E. coordinated the project, provided resources and contributed to the writing of the manuscript.

## Competing interests
All authors declare no competing interests.
