## [Peer Review File · Communications Chemistry]

Reviewers' comments:

Reviewer #1 (Remarks to the Author):

In the manuscript "Microwave-assisted synthesis of Metal Organic Chalcogenolate Assemblies (MOCHAs) as electrocatalysts for Syngas production" the authors test two novel materials for the catalysed production of Syngas from CO₂ reduction. The authors further claim to develop a novel synthetic strategy for the production of AgSePh and AgSPh reducing the reaction time needed and improving the yield 100-fold. The novelty of the use of this kind of materials for CO₂ reduction is very relevant and of broad interest for the chemistry community. In fact, is it not obvious at all that a well-passivated AgSe or AgS layer by organic molecules would provide catalytic activity such to be also relevant in a technologically very impactful chemical process. Nevertheless, I think that the claim should be better contextualized. The authors should report the state of the art material performances for this kind of reaction process in terms of faradaic efficiency, stability and rate of CO and H₂ production. Such that the reader can better appreciate the potential impact of the new materials presented here. Some other concerns I have:

- The authors claim that their synthesis "shorten the synthesis time from 3 days to only 1-5 hours and increases the yield of the material by a factor of ~100, thus allowing extended stability studies on MOCHAs." I would like to point out that ref 63 reports 100% yield for 30 min reaction, and in the article <https://pubs.acs.org/doi/full/10.1021/jacs.1c09106> it is reported that "1:1 ratio of Ph₂Se₂:AgNO₃ was needed to obtain 100% formation yield of AgSePh". I think the authors should focus more on the quantities obtained compared to the yield and I ask them to justify the "extended stability" claim.

In fact, the sentence "So far, MOCHAs have been synthesized through a biphasic process using an at 80°C⁵²" (page 2) is incomplete as a sentence and in its content meaning, please refer to refs [63] and to <https://pubs.acs.org/doi/full/10.1021/jacs.1c09106>. In general, the presented synthesis should be compared to the syntheses reported by Hohman, Tisdale and Caironi. In view of this also the conclusions should be toned down.

- Page 3: "Taking a molecular weight of 264 g mol⁻¹ for [AgSePh]_∞, this amounts to a greatly improved yield of 83%⁹" the reference is wrong and the number is not explained. How the authors got it?

- Page 6: I think the authors should better explain the experiment where CO₂ reaction yields CO and H₂, where the latter come from? Please integrate the text with an explicit reaction. Consequently I cannot understand this sentence in page 8: "In both cases silver chalcogenides favour the formation of CO more than the syngas. However, Mithrene and Thiorene prefer syngas formation which makes them more suitable for roomtemperature formation of syngas." Please clarify.

- The literature is quite complete but I recommend the integration of relevant literature concerning the synthesis and optical studies of MOCHAs, recently published by the Tisdale group at MIT. (<https://pubs.acs.org/doi/full/10.1021/jacs.1c09106> ; <https://pubs.acs.org/doi/full/10.1021/acsnano.1c07498>) and by Maserati et al (<https://pubs.rsc.org/en/content/articlehtml/2020/mh/c9mh01917k> ; <https://pubs.rsc.org/en/content/articlehtml/2021/nr/d0nr07409h>)

For the reasons discussed above, I think the impact of the novel synthesis should be better discussed in the literature context and the claims of impact toned down for what concerns the synthesis.

I also do not understand the need of using made up names like Mithrene when there is a chemical formula (AgSePh), and a chemical name (Silver phenylselenolate or benzeneselenolate) already reported by the first group that developed the compound in 2002.

The methodologies are clear and but the authors should provide better description of the reaction for assuring the reproduction of the presented experiments. In addition, I think the scholar presentation must be improved. I point out some corrections:

- Page 2: "Further a MOCHA film bound to the substrate is obtained via this method, making MOCHA isolation difficult." I don't understand "further" and I object that actually in this way the isolation is easy because the metal is fully reacted to become MOCHA.
- Page 2: "Crystallographic and optical studies on MOCHAs in particular on the silver benzeneselenolate MOCHA "Mithrene" have been carried out since then 53,55,57,58,60. However, these methods yield quantities amounting to 1 mg which in return hinders further practical applications." Which methods? The clarity of presentation should be improved.
- Page 3: "Thiorenene on the other hand, does not exhibit a delocalization of excitons in two dimensions due to its linear network, causing a lack of emission in the visible range for Thiorenene" please revise the sentence.
- Page 5: "Stability tests upon light illumination with sources of different wavelength (365nm, 456nm, Solar Simulator) showed that Mithrene powder turned from a bright yellow colour into a darker brownish-yellow upon longer illumination". Longer compared to what time?
To conclude, I think the manuscript need a deep revision, but it is of interest for Nature Communication Chemistry.

Reviewer #2 (Remarks to the Author):

In the submitted article, the authors seek alternative methods to synthesize catalytic material for syngas production (mixture of H₂ and CO). For this purpose, they proposed alleviating the synthesis of metal organic chalcogenolates by using a microwave reactor instead of the more energetically expensive methods. They were able to accomplish this and prove that they serve as a decent electrocatalytic system for the formation of syngas from CO₂. The work looks relevant to the world need to find alternative method of production of raw material for fuel and it seems to suggest the production of MOCHAs is a potential suitor for this. I would recommend for publication with some minor reviews.

First, some questions:

- 1) Is there data on how these compare to the state-of-the-art catalytic system in syngas production
- 2) These material seem thin enough for TEM, any particular reason these were not evaluated in TEM? Are they stable under the electron beam? I can see what it seems to be some charging in the SEM. I am asking because it would be interesting to see in more detail the before and after electrocatalysis by TEM

Second, some comments:

- 1) In many of the figures in the supplementary information, the legend for the graph is blocking part of the data (this happens in the IR and XRD), even though there may not be relevant features, it is good practice and necessary to show all the data as is. Please revise the figures in the supplementary so that no data is blocked. This is also true for the XRD in figure 3.
- 2) The EDS and the XPS measurement need to be shown. It is ok to summarize the data in a table, but the it is important to show the XPS/EDS data.
- 3) In page 2, paragraph 2, please revise Ag₂O to read Ag₂.
- 4) Please revise the reference format, since there's inconsistent formatting.

Dear referees,

Thank you very much for reviewing our manuscript and making some excellent critical points. We have used the comments to strengthen the manuscript and have implemented the suggestions in the revised text that we are resubmitting. We colour the referee comments in **RED**, our response in **BLUE** and the text added to the manuscript in **GREEN**. The text we have added to the manuscript is highlighted in yellow.

Thank you very much in advance for consideration of our revised work.

Kind regards,
Dr. Dogukan H. Apaydin

Reviewer: 1

In the manuscript “Microwave-assisted synthesis of Metal Organic Chalcogenolate Assemblies (MOCHAs) as electrocatalysts for Syngas production” the authors test two novel materials for the catalysed production of Syngas from CO₂ reduction. The authors further claim to develop a novel synthetic strategy for the production of AgSePh and AgSPh reducing the reaction time needed and improving the yield 100-fold. The novelty of the use of this kind of materials for CO₂ reduction is very relevant and of broad interest for the chemistry community. In fact, is it not obvious at all that a well-passivated AgSe or AgS layer by organic molecules would provide catalytic activity such to be also relevant in a technologically very impactful chemical process. Nevertheless, I think that the claim should be better contextualized. The authors should report the state-of-the-art material performances for this kind of reaction process in terms of faradaic efficiency, stability and rate of CO and H₂ production. Such that the reader can better appreciate the potential impact of the new materials presented here.

We are grateful to the reviewer for the appreciation of our work. We believe that we managed to address the reviewer’s concerns below.

Some other concerns I have:

- The authors claim that their synthesis “shorten the synthesis time from 3 days to only 1-5 hours and increases the yield of the material by a factor of ~100, thus allowing extended stability studies on MOCHAs.” I would like to point out that ref 63 reports 100% yield for 30 min reaction, and in the article <https://pubs.acs.org/doi/full/10.1021/jacs.1c09106> it is reported that “1:1 ratio of Ph₂Se₂:AgNO₃ was needed to obtain 100% formation yield of AgSePh”. I think the authors should focus more on the quantities obtained compared to the yield and I ask them to justify the “extended stability” claim.

Thank you for this point. Indeed, the word “extended” does not fit here as there is no comparison in the literature. We changed the sentence to read:

“Furthermore, we evaluated the performance of [AgSePh]_∞ and [AgSPh]_∞ for electrocatalytic syngas production. In addition to catalytic studies we perform for the first time long-term stability studies of MOCHAs in various environments.”

In fact, the sentence “So far, MOCHAs have been synthesized through a biphasic process using an at 80°C⁵²” (page 2) is incomplete as a sentence and in its content meaning, please refer to refs [63] and to <https://pubs.acs.org/doi/full/10.1021/jacs.1c09106>. In general, the presented synthesis should be compared to the syntheses reported by Hohman, Tisdale and Caironi. In view of this also the conclusions should be toned down.

We appreciate the constructive comments of the reviewer. We have now altered the text to read as follows:

“So far, different synthesis methods for MOCHAs did not result in high yields which hinder further use of MOCHAs in applications.”

- Page 3: “Taking a molecular weight of 264 g mol⁻¹ for [AgSePh]_∞, this amounts to a greatly improved yield of 83%⁹” the reference is wrong, and the number is not explained. How the authors got it?

We thank the reviewer for the keen eye. Indeed, our reference was wrong. We have now cited the correct reference and changed the sentence accordingly to read:

“Taking a molecular weight of 264 g mol⁻¹ for [AgSePh]_∞, as suggested by Schriber et al. This amounts to a greatly improved yield of 83%.”

- Page 6: I think the authors should better explain the experiment where CO₂ reaction yields CO and H₂, where the latter come from? Please integrate the text with an explicit reaction. Consequently, I cannot understand this sentence in page 8:” In both cases silver chalcogenides favour the formation of CO more than the syngas. However, Mithrene and Thiorene prefer syngas formation which makes them more suitable for room temperature formation of syngas.” Please clarify.

We introduced the following text into the manuscript to make the origins of CO and H₂ in a CO₂ reduction clear:

“Electrocatalytic CO₂RR in aqueous environment is accompanied by the hydrogen evolution reaction (HER) due to the simultaneous reduction of water (or protons H⁺) and CO₂. The following two reactions occur²⁹:

By producing Syngas electrocatalytically, the drawback of a competing reaction can be bypassed and the HER fully integrated in the use. Another advantage of electrocatalytic Syngas

production compared to other Syngas production techniques including solid oxide electrolysis cells (SOECs), steam reforming and thermocycles is its room temperature application^{1,2,13,27}. Electrocatalytic syngas production is thus a cost effective, environmentally friendly alternative for producing Syngas. However, the finding and development of cheap and suitable catalysts is still a huge challenge in the field. A summary of potential catalysts and their Faraday Efficiencies, current densities and experimental parameters is summarized in Table 1. and compared to the results obtained with MOCHA/CP electrodes.”

- The literature is quite complete, but I recommend the integration of relevant literature concerning the synthesis and optical studies of MOCHAs, recently published by the Tisdale group at MIT. (<https://pubs.acs.org/doi/full/10.1021/jacs.1c09106> ; <https://pubs.acs.org/doi/full/10.1021/acsnano.1c07498>) and by Maserati et al (<https://pubs.rsc.org/en/content/articlehtml/2020/mh/c9mh01917k> ; <https://pubs.rsc.org/en/content/articlehtml/2021/nr/d0nr07409h>)

We thank the reviewer for the literature suggestions. We have now implemented these as references: 61-64.

For the reasons discussed above, I think the impact of the novel synthesis should be better discussed in the literature context and the claims of impact toned down for what concerns the synthesis.

We believe we have now addressed the impact of our synthesis better in terms of high yields and toned down our claims.

I also do not understand the need of using made up names like Mithrene when there is a chemical formula (AgSePh), and a chemical name (Silver phenylselenolate or benzeneselenolate) already reported by the first group that developed the compound in 2002. The methodologies are clear, but the authors should provide better description of the reaction for assuring the reproduction of the presented experiments. In addition, I think the scholar presentation must be improved. I point out some corrections:

We have now changed the words mithrene and thiorene to appropriate chemical notations.

- Page 2: “Further a MOCHA film bound to the substrate is obtained via this method, making MOCHA isolation difficult.” I don’t understand “further” and I object that actually in this way the isolation is easy because the metal is fully reacted to become MOCHA.

We agree with the reviewer that this sentence can lead to confusion. We have now altered the sentence to read:

“A MOCHA film bound to the substrate obtained via this method, limits the use of various substrates for growing MOCHA films. In addition, such metal thin films require a physical vapor deposition setup which is not a common laboratory infrastructure.”

- Page 2: “Crystallographic and optical studies on MOCHAs in particular on the silver benzeneselenolate MOCHA “Mithrene” have been carried out since then 53,55,57,58,60. However, these methods yield quantities amounting to 1 mg which in return hinders further practical applications.” Which methods? The clarity of presentation should be improved.

We have altered this sentence (and updated the references as requested by the reviewer) to have a clear meaning. Now the sentence reads:

“Crystallographic and optical studies on MOCHAs in particular on the silver benzeneselenolate MOCHA “*Mithrene*” have been carried out since then ^{53,55,57,58,60,62–65}. However, either the quantities are very low (biphasic synthesis) or MOCHA film thickness (tarnishing method) does not exceed 20nm. Such limitations seem to hinder further applications of MOCHAs.”

- Page 3: “Thiorene on the other hand, does not exhibit a delocalization of excitons in two dimensions due to its linear network, causing a lack of emission in the visible range for Thiorene” please revise the sentence.

We have revised the sentence to make the explanation clearer by introducing “Ag-Ag linear chains” into it.

- Page 5: “Stability tests upon light illumination with sources of different wavelength (365nm, 456nm, Solar Simulator) showed that Mithrene powder turned from a bright yellow colour into a darker brownish-yellow upon longer illumination”. Longer compared to what time?

Thank you for noticing this. We have now altered the sentence to read:

“Stability tests upon light illumination with sources of different wavelength (365nm, 456nm, Solar Simulator) showed that [AgSePh]_∞ powder turned from a bright yellow colour before illumination into a darker brownish-yellow coloured powder upon illumination times longer than 24h”

To conclude, I think the manuscript need a deep revision, but it is of interest for Nature Communication Chemistry.

Reviewer: 2

In the submitted article, the authors seek alternative methods to synthesize catalytic material for syngas production (mixture of H₂ and CO). For this purpose, they proposed alleviating the synthesis of metal organic chalcogenolates by using a microwave reactor instead of the more energetically expensive methods. They were able to accomplish this and prove that they serve

as a decent electrocatalytic system for the formation of syngas from CO₂. The work looks relevant to the world need to find alternative method of production of raw material for fuel and it seems to suggest the production of MOCHAs is a potential suitor for this. I would recommend for publication with some minor reviews.

First, some questions:

1) Is there data on how these compare to the state-of-the-art catalytic system in syngas production

Thank you very much for pointing this out. We have now added a table on page 8 of the manuscript summarizing the state-of-the-art catalytic systems for syngas production:

Catalyst	Potential range (Vvs. RHE)	Current density (mA cm ⁻²)	CO:H ₂ ratio	FE _{CO} (%)	FE _{Total} (%)	Electrolyte	Refs.
Pd/C	-0.5 ~ -1.0	0.6 (-0.7 V)	0.5-2.0	60	90	0.5M NaHCO ₃	68
Zn/Cu foam	-0.6 ~ -1.3	40 (-1.3 V)	0.2-2.31	40	85	0.5M KHCO ₃	69
Au-NPs	-0.6 ~ -1.3	/	0.5-1.0	45	100	0.1M KHCO ₃	70
Fe/FeN ₄ C	-0.45 ~ -0.8	39.33 (-0.8 V)	0.1-0.9	52	100	0.5M KHCO ₃	71
MoSeS alloy	-0.6 ~ -1.6	43 (-1.15 V)	1.0	45.2	96	EMIM-BF ₄ solution	47
This work							
[AgSePh] _∞	-0.8~ -1.0	1.6 - 6.3	0-1.2	5-55	89-100	0.5M KHCO ₃	/
[AgSPh] _∞	-0.8~ -1.0	3.3 - 7.9	0.5-1.6	34-54	88-98	0.5M KHCO ₃	/

2) These material seem thin enough for TEM, any particular reason these were not evaluated in TEM? Are they stable under the electron beam? I can see what it seems to be some charging in the SEM. I am asking because it would be interesting to see in more detail the before and after electrocatalysis by TEM.

We have attempted to take TEM images of these crystals. We used crystals from the same synthesis batch as in SEM images. However, under electron beam in TEM the crystals seem misshaped for a short time before the beam destroys them. So far, we have not been successful obtaining a TEM image of MOCHAs we have synthesized.

Second, some comments:

1) In many of the figures in the supplementary information, the legend for the graph is blocking part of the data (this happens in the IR and XRD), even though there may not be relevant features, it is good practice and necessary to show all the data as is. Please revise the figures in the supplementary so that no data is blocked. This is also true for the XRD in figure 3.

We thank the reviewer for the keen eye. We have now updated all the legends in figures. Now all the data points are visible.

2) The EDS and the XPS measurement need to be shown. It is ok to summarize the data in a table, but the it is important to show the XPS/EDS data.

Both XPS and EDS graphs have been added to the Supporting information. We also took this opportunity to perform another set of XPS measurements as AgSePh measurement did not work out the last time we measured it. Now both MOCHAs have XPS measurements performed and the Table 1 in supporting information is updated accordingly.

3) In page 2, paragraph 2, please revise Ag₂O to read Ag₂.

To keep it unified with the previous sentence we have now changed it to “silver oxide”

4) Please revise the reference format, since there's inconsistent formatting.

Now the reference format should be unified.

REVIEWERS' COMMENTS:

Reviewer #1 (Remarks to the Author):

Dear Editor,

I am satisfied with the corrections the authors made and I think the manuscript should be accepted as is for publication.

Reviewer #2 (Remarks to the Author):

The authors answers to all my comments was satisfactory. Publish as is.